# Use of gentamicin-collagen sponge (Collatamp® G) in minimally invasive colorectal cancer surgery: A propensity score-matched study

**Kil-yong Lee, Jaeim Lee\***, **Youn Young Park, Seong Taek Oh**

Division of Coloproctology, Department of Surgery, Uijeongbu St. Mary's Hospital, College of Medicine, The Catholic University of Korea, Uijeongbu-si, South Korea

\* lji96@catholic.ac.kr

## Abstract

### Background

Minimally invasive surgery is commonly used to treat patients with colorectal cancer, although it can cause surgical site infections (SSIs) that can affect the oncologic outcome. Use of a gentamicin-collagen sponge may help reduce the occurrence of SSIs. We aimed to determine the effectiveness of a gentamicin-collagen sponge in reducing SSIs in minimally invasive surgery for colorectal cancer.

### Methods

We retrospectively reviewed the records of 310 patients who were diagnosed with colorectal cancer at our hospital and underwent minimally invasive surgery between December 1, 2018, and February 28, 2021. Propensity score matching was conducted with a 1:1 ratio using logistic regression. The primary outcome was the incidence of SSIs in the mini-laparotomy wound. The secondary endpoints were factors affecting the incidence of SSIs.

### Results

After propensity score matching, 130 patients were assigned to each group. There were no differences in clinical characteristics between the two groups. SSIs occurred in 2 (1.5%) and 3 (2.3%) patients in the gentamicin-collagen sponge and control groups, respectively (p<0.999). The following factors showed a statistically significant association with SSIs: body mass index >25 kg/m$^2$ (odds ratio, 39.0; 95% confidence interval, 1.90–802.21; p = 0.018), liver disease (odds ratio, 254.8; 95% confidence interval, 10.43–6222.61; p = 0.001), and right hemicolectomy (odds ratio, 36.22; 95% confidence interval, 2.37–554.63; p = 0.010).

**Data Availability Statement:** Our data from this study are available upon request owing to the restrictions (data contain potentially identifying information [birth dates]) by the Institutional

Review Board of Uijeongbu St. Mary's Hospital, The Catholic University of Korea, South Korea. Please contact our IRB committee for data access via e-mail (irbujb@catholic.ac.kr).

**Funding:** The authors received no specific funding for this work.

**Competing interests:** The authors have declared that no competing interests exist.

## Conclusion

Applying a gentamicin-collagen sponge to the mini-laparotomy wound did not reduce the frequency of SSIs. Further studies should be conducted on whether the selective use of gentamicin-collagen sponges may help reduce SSIs in high-risk patients.

## Introduction

In patients with colorectal cancer, minimally invasive surgery reduces the hospital stay after surgery and increases patient satisfaction by reducing the incision area. A recent meta-analysis showed that minimally invasive surgery decreased the frequency of surgical site infections (SSIs) from 8.0% to 5.8% compared to open surgery (risk ratio: 0.72, 95% confidence interval [CI] 0.60–0.88) [1]; however, minimally invasive techniques cannot definitively prevent SSIs in colorectal as these are clean-contaminated operations.

The effectiveness of the gentamicin-collagen sponge in reducing SSIs has been reported in various fields of surgery, such as thoracic and orthopedic surgery [2, 3]. However, in the colorectal surgery field, a large-scale randomized control study [4] of gentamicin-containing sponges failed to prove their effectiveness. Nevertheless, a recent meta-analysis that excluded this study due to high risk of bias reported that, based on sensitivity analyses of abdominal wounds, gentamicin-collagen sponges could reduce the risk of SSI (relative risk [RR], 0.38; 95% CI, 0.20–0.72) [5].

For colorectal cancer patients, preventing SSI is important as they can affect long-term survival [6]. Few studies on whether gentamicin-collagen sponges (especially Collatamp® G (Schering-Plough, Stockholm, Sweden)) can prevent SSIs in laparoscopic colorectal cancer surgery have been performed. Therefore, we aimed to investigate the incidence of SSI after laparoscopic colorectal cancer surgery when using the Collatamp.

## Materials and methods

This study was approved by the institutional review board (IRB) of the Catholic University of Korea (IRB number: UC21RISI0027). The study was performed in accordance with the relevant guidelines and regulations of the IRB. The investigation conformed with the principles outlined in the Declaration of Helsinki of 1964. Informed consent for participation was waived under IRB approval from the institutional review board of the Catholic University of Korea.

### Patients

We enrolled patients who were diagnosed with colorectal cancer at our hospital and underwent minimally invasive surgery from December 1, 2018, to February 28, 2021. The prospectively collected database was analyzed retrospectively. Patients who underwent primary tumor resection via a laparoscopic or robotic approach were included in the study. The inclusion criteria were laparoscopic or robotic operations for biopsy-proven colorectal cancer and specimen extraction via mini-laparotomy wounds. The exclusion criteria were as follows: (1) open surgery, including conversion from laparoscopy; (2) transanal local resection or abdominoperineal resection for rectal cancer; (3) Hartmann's operation; (4) laparoscopic biopsy only; and (5) early postoperative mortality within 7 days.

## Procedure

The bowel was prepared using a polyethylene glycol electrolyte solution (4L; CoLyte; Taejoon Pharma Co., Ltd, Seoul, Korea) if the patient had no signs of complete obstruction or perforation, and oral antibiotics for bowel preparation had not been administered before surgery. One hour preoperatively, intravenous cefoxitin 2 g was administered for prophylaxis against infection. Specimen extraction or anastomosis was performed with an additional mini-laparotomy of approximately 5 cm for all patients. For right and left hemicolectomy, an extracorporeal anastomosis was performed using a mini-laparotomy in the upper midline and left upper quadrant, respectively. For an anterior or low anterior resection, after performing an intracorporeal rectal transection, a transverse incision was created in the left lower quadrant to extract the colon and resect it with appropriate margins followed by an intracorporeal end-to-end anastomosis. Dual-ring wound protectors were used for all mini-laparotomy wounds. After closing the abdominal wall fascia, a gentamicin-collagen sponge [Collatamp®G (Schering-Plough, Stockholm, Sweden); 5 cm × 5 cm, containing 50 mg gentamicin] was inserted in the subcutaneous layer.

## Definitions

In our study, we focused on SSIs at the mini-laparotomy wound because we aimed to determine the effectiveness of the Collatamp® G in preventing SSIs. An SSI was defined as a clinically reported infection of the mini-laparotomy wound occurring within 30 days of the surgery according to the Center for Disease Control and Prevention (CDC) guidelines [7].

Liver disease was defined as the presence of hepatitis B or C, or any form of liver cirrhosis.

Sealed-off perforation was defined as a perforation with a localized abscess on the preoperative computed tomography image or as an intraoperative field without free perforation (i.e., fecal contamination or dirty fluid collection in the peritoneal cavity). Microperforation was defined as postoperative pathologic findings of a perforation.

Partial obstruction was defined as inability of the colonoscope to enter an encircling lesion in a patient who could pass stool. Complete obstruction was defined as no stenting or inability to place a stent.

Progression-free survival was defined as the time from the date of surgery to the date of a diagnosis of recurrence, cancer progression, or death from any cause. The date of the last outpatient visit to the doctor in charge was the last follow-up day for progression-free survival. Overall survival was defined as the time from the date of the operation to the date of death from cancer or any cause. The last follow-up day for overall survival was the last outpatient visit to our hospital.

## Outcomes

The primary outcome was the incidence of SSI in the mini-laparotomy wound. The secondary outcome were the factors that affected the development SSIs.

## Statistical analysis

For comparisons between the two groups, categorized variables were analyzed using Fisher's exact test, the chi-square test, and linear-by-linear association; while continuous variables were analyzed using the Mann–Whitney test and the Student t-test. Categorized variables were expressed as numbers and percentages. Continuous variables were expressed as mean ± standard deviation. To analyze the survival in the two groups, Kaplan–Meier curves with the log-rank test were used.

Propensity-scored matching with a 1:1 ratio, using logistic regression with the nearest-neighbor method, was conducted to match the two groups. Propensity score matching was conducted using the R package MatchIt (R version 3.2.2; R Foundation for Statistical Computing, Vienna, Austria) [8]. The variables included in the matching were age, sex, height, weight, body mass index (BMI), underlying diseases (e.g., diabetes, hypertension, cardiac disease, pulmonary disease, liver disease, cerebrovascular disease), American Society of Anesthesiologists physical status classification, smoking, alcohol use, cancer location, operation name, operation type (i.e., laparoscopy or robotic), combined resection, preoperative obstruction, preoperative perforation, emergency operation, preoperative hemoglobin level, preoperative albumin level, packed red blood cell transfusion (i.e., preoperative or intraoperative), tumor stage, T stage, N stage, and M stage.

For multivariable analysis of factors affecting SSIs, logistic regression with backward stepwise selection of factors with a p-value <0.2 in the univariable analysis was performed as previously described [9, 10]. SPSS v.21 (IBM Corporation, New York, NY, USA) was used to conduct the analysis. Differences with a p-value <0.5 were considered statistically significant.

## Results

After excluding 75 patients who underwent open surgery, 343 patients who underwent minimally invasive surgery from December 1, 2018, to February 28, 2021 remained. Patients who had undergone abdominoperineal resection, a Hartmann operation, or laparoscopic biopsy were excluded. Among the remaining 312 patients whose specimens were extracted through the mini-laparotomy site, 2 were excluded because they died within 7 days after surgery. Therefore, 310 patients were ultimately analyzed for the development of SSIs (Fig 1). One hundred and thirty patients and 180 patients were in the Collatamp group and the control group, respectively. The clinical characteristics of the two groups are shown in Table 1. In terms of patient characteristics, there were significantly more cases of cardiac disease in the control group before propensity score-matching; however, as with most other characteristics, there was no difference in the rates of SSIs.

After propensity score matching was conducted by correcting for covariables that affected the development SSI, 130 patients were assigned to each group. The two groups showed no significant differences in clinical characteristics (Table 2). SSIs occurred in 2 (1.5%) and 3 (2.3%) patients in the Collatamp and control groups, respectively, showing no statistically significant difference (p >0.999). The median length of hospital stay in the Collatamp and control groups was 6.7 days and 6.5 days, respectively, which was not significantly different (p = 0.568). The incidence of postoperative complications, based on the Clavien-Dindo classification, was not significantly different (p = 0.546). Over an average follow-up period of 324 days, the estimated 2-year progression-free survival was higher in the Collatamp group (92.2%) than in the control group (77.3%); however, the difference was not significant (log-rank p-value = 0.092) (Fig 2). Similarly, over an average follow-up period of 347 days, there was no significant difference in estimated 2-year overall survival between the Collatamp group (94.8%) and the control group (92.7%; log-rank p-value = 0.581) (Fig 3).

The univariable analysis for factors affecting SSI among all 310 patients revealed that liver disease (p = 0.004), perforation (p<0.001), and right hemicolectomy (p = 0.001) were all associated with the development of SSI (Table 3).

The multivariable analysis of these factors revealed that BMI >25 kg/m$^2$ [OR, 39.0; 95% confidence interval (CI), 1.90–802.21; p = 0.018], liver disease (OR, 254.8; 95% CI, 10.43–6222.61; p = 0.001), and right hemicolectomy (OR, 36.22; 95% CI, 2.37–554.63; p = 0.010) were independently associated with SSI (Table 4).

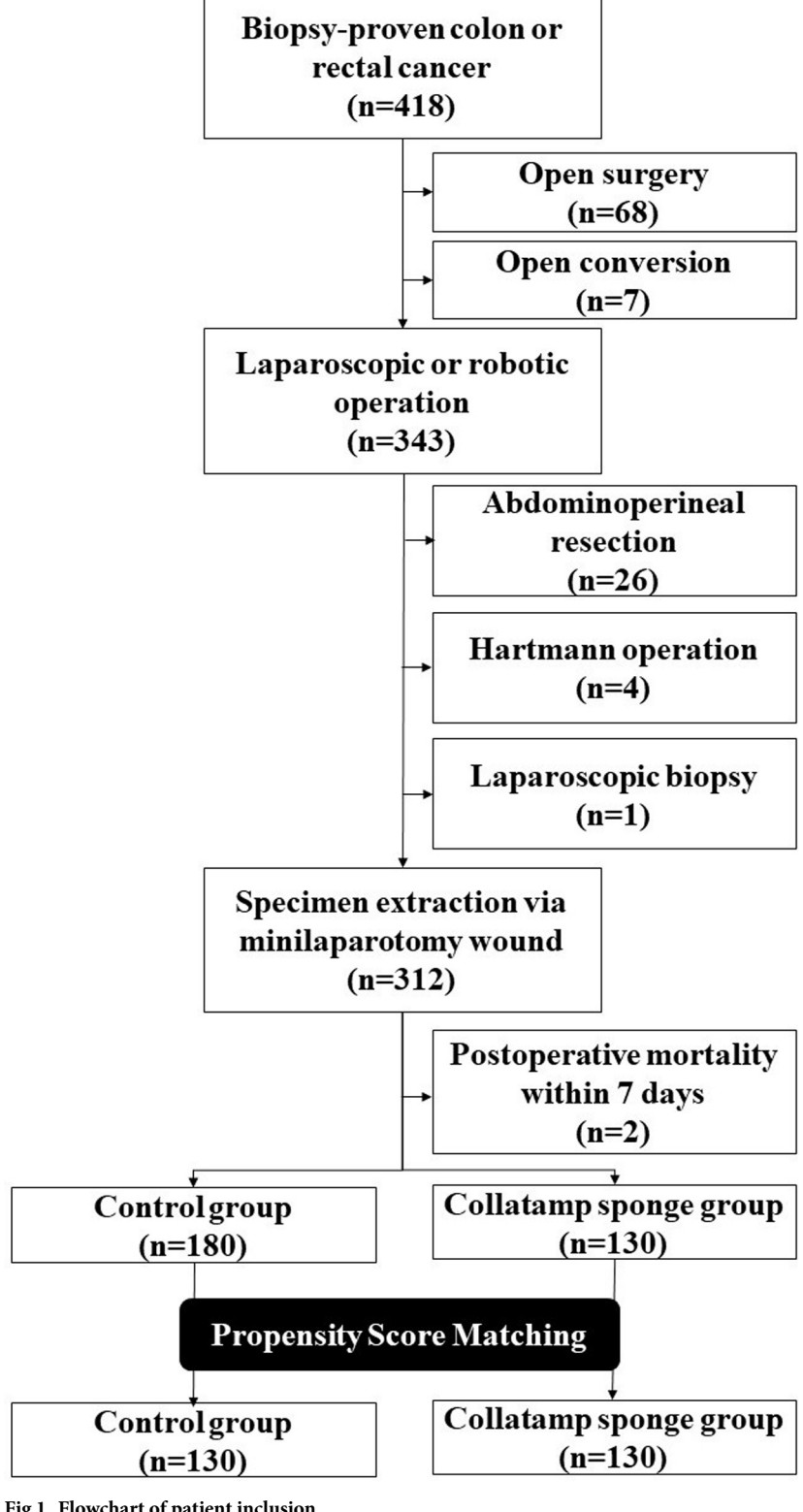

**Fig 1. Flowchart of patient inclusion.**

**Table 1. Patients' characteristics before propensity score-matching.**

| | Control group (n = 180) | Collatamp sponge group (n = 130) | p-value |
|---|---|---|---|
| **Age (y)** | 67.7 ± 11.9 | 66.8 ± 11.6 | 0.503 |
| **Sex** | | | 0.755 |
| Male | 109 (60.6%) | 81 (62.3%) | |
| Female | 71 (39.4%) | 49 (37.7%) | |
| **Height (cm)** | 162.0 ± 9.2 | 160.9 ± 8.4 | 0.273 |
| **Weight (kg)** | 61.6 ± 11.8 | 61.6 ± 11.2 | 0.987 |
| **Body mass index (kg/m$^2$)** | 23.4 ± 3.6 | 23.7 ± 3.7 | 0.356 |
| **Underlying disease** | | | |
| Hypertension | 101 (56.1%) | 77 (59.2%) | 0.584 |
| Diabetes | 56 (31.1%) | 37 (28.5%) | 0.615 |
| Cardiac disease | 31 (17.2%) | 9 (6.9%) | 0.008 |
| Pulmonary disease | 20 (11.1%) | 8 (6.2%) | 0.133 |
| Liver disease | 11 (6.1%) | 3 (2.3%) | 0.112 |
| Cerebrovascular disease | 19 (10.6%) | 14 (10.8%) | 0.952 |
| Chronic kidney disease | 12 (6.7%) | 8 (6.2%) | 0.856 |
| **ASA classification** | | | 0.365 |
| 1 | 16 (8.9%) | 10 (7.7%) | |
| 2 | 123 (68.3%) | 99 (76.2%) | |
| 3 | 41 (22.8%) | 21 (16.2%) | |
| **Smoking** | | | 0.348 |
| Ex-smoker | 22 (12.2%) | 13 (7.7%) | |
| Current smoker | 27 (15.0%) | 17 (13.1%) | |
| **Alcohol** | | | <0.001 |
| Ex-alcoholic | 2 (1.1%) | 2 (1.5%) | |
| Current alcoholic | 31 (17.2%) | 5 (3.8%) | |
| **Hemoglobin (g/dL)** | 12.0 ± 2.4 | 12.0 ± 2.3 | 0.875 |
| **Albumin (g/dL)** | 4.0 ± 0.6 | 3.9 ± 0.5 | 0.087 |
| **Cancer location** | | | 0.215 |
| Cecum | 5 (2.8%) | 2 (1.5%) | |
| Appendix | 1 (0.6%) | 0 | |
| Ascending colon | 29 (16.1%) | 28 (21.5%) | |
| Hepatic flexure | 8 (4.4%) | 9 (6.9%) | |
| Transverse colon | 7 (3.9%) | 12 (9.2%) | |
| Splenic flexure | 4 (2.2%) | 2 (1.5%) | |
| Descending colon | 2 (1.1%) | 5 (3.8%) | |
| Sigmoid colon | 44 (24.4%) | 29 (22.3%) | |
| Rectosigmoid | 13 (7.2%) | 4 (3.1%) | |
| Rectum | 65 (36.1%) | 38 (29.2%) | |
| Double primary | 2 (1.1%) | 1 (0.8%) | |
| **Obstruction** | | | 0.023 |
| None | 122 (67.8%) | 104 (80.0%) | |
| Partial | 29 (16.1%) | 14 (10.8%) | |
| Stent insertion | 24 (13.3%) | 10 (7.7%) | |
| Complete obstruction | 5 (2.8%) | 2 (1.5%) | |
| **Perforation** | | | 0.505 |
| None | 165 (91.7%) | 123 (94.6%) | |
| Microperforation | 11 (6.1%) | 6 (4.6%) | |
| Sealed-off perforation | 4 (2.2%) | 1 (0.8%) | |

(*Continued*)

**Table 1.** (Continued)

| | Control group (n = 180) | Collatamp sponge group (n = 130) | p-value |
|---|---|---|---|
| **Operation methods** | | | 0.086 |
| Right hemicolectomy | 45 (25.1%) | 51 (39.2%) | |
| Left hemicolectomy | 7 (3.9%) | 6 (4.6%) | |
| Anterior resection | 70 (39.1%) | 33 (25.4%) | |
| Low anterior resection | 55 (30.7%) | 38 (29.2%) | |
| Total or subtotal colectomy | 2 (1.1%) | 2 (1.5%) | |
| **Combined resection** | | | 0.898 |
| No | 170 (94.4%) | 122 (93.8%) | |
| Liver | 1 (0.6%) | 2 (1.5%) | |
| Uterus | 2 (1.1%) | 0 | |
| Small bowel | 1 (0.6%) | 2 (1.5%) | |
| Stomach. | 1 (0.6%) | 0 | |
| Gallbladder | 1 (0.6%) | 0 | |
| Ovary | 2 (1.1%) | 3 (2.3%) | |
| Spleen | 1 (0.6%) | 0 | |
| Bladder | 1 (0.6%) | 1 (0.8%) | |
| **Operation type** | | | 0.078 |
| Laparoscopic | 168 (93.3%) | 127 (97.7%) | |
| Robotic | 12 (6.7%) | 3 (2.3%) | |
| **Emergency operation** | 8 (4.4%) | 4 (3.1%) | 0.538 |
| **Transfusion** | | | |
| Preoperative | 17 (9.4%) | 13 (10.0%) | 0.870 |
| Intraoperative | 25 (13.9%) | 9 (6.9%) | 0.053 |
| **Stage** | | | 0.362 |
| 0 (Tis) | 6 (3.3%) | 2 (1.5%) | |
| 1 | 45 (25.0%) | 31 (23.8%) | |
| 2 | 41 (22.8%) | 27 (20.8%) | |
| 3 | 62 (34.4%) | 49 (37.7%) | |
| 4 | 26 (14.4%) | 21 (16.2%) | |
| **T stage** | | | 0.539 |
| Tis | 6 (3.3%) | 2 (1.5%) | |
| 1 | 27 (15.0%) | 20 (15.4%) | |
| 2 | 21 (11.7%) | 15 (11.5%) | |
| 3 | 97 (53.9%) | 72 (55.4%) | |
| 4a | 28 (15.6%) | 17 (13.1%) | |
| 4b | 1 (0.6%) | 4 (3.1%) | |
| **N stage** | | | 0.914 |
| 0 | 95 (52.8%) | 64 (49.2%) | |
| 1 | 45 (25.0%) | 43 (33.1%) | |
| 2 | 40 (22.2%) | 23 (17.7%) | |
| **M stage** | | | 0.782 |
| 0 | 153 (85.0%) | 109 (83.8%) | |
| 1 | 27 (15.0%) | 21 (16.2%) | |
| **Surgical site infection** | 6 (3.3%) | 2 (1.5%) | 0.475 |

ASA, American Society of Anesthesiologists

**Table 2. Patients' characteristics after propensity score matching.**

| | Control group (n = 130) | Collatamp sponge group (n = 130) | p-value |
|---|---|---|---|
| **Age (y)** | 66.7 ± 12.0 | 66.8 ± 11.6 | 0.966 |
| **Sex** | | | 0.703 |
| Male | 78 (60.0%) | 81 (62.3%) | |
| Female | 52 (40.0%) | 49 (37.7%) | |
| **Height (cm)** | 161.3 ± 8.8 | 160.9 ± 8.4 | 0.716 |
| **Weight (kg)** | 61.4 ± 10.6 | 61.6 ± 11.2 | 0.874 |
| **BMI (kg/m$^2$)** | 23.5 ± 3.3 | 23.7 ± 3.7 | 0.638 |
| **Underlying disease** | | | |
| Hypertension | 72 (55.4%) | 77 (59.2%) | 0.531 |
| Diabetes | 36 (27.7%) | 37 (28.5%) | 0.890 |
| Cardiac disease | 17 (13.1%) | 9 (6.9%) | 0.098 |
| Pulmonary disease | 8 (6.2%) | 8 (6.2%) | >0.999 |
| Liver disease | 5 (3.8%) | 3 (2.3%) | 0.722 |
| Cerebrovascular disease | 15 (11.5%) | 14 (10.8%) | 0.844 |
| Chronic kidney disease | 5 (3.8%) | 8 (6.2%) | 0.393 |
| **ASA classification** | | | 0.903 |
| 1 | 13 (10.0%) | 10 (7.7%) | |
| 2 | 92 (70.8%) | 99 (76.2%) | |
| 3 | 25 (19.2%) | 21 (16.2%) | |
| **Smoking** | | | 0.488 |
| Ex-smoker | 14 (10.8%) | 13 (7.7%) | |
| Current smoker | 19 (14.6%) | 17 (13.1%) | |
| **Alcohol use** | | | 0.131 |
| Ex-alcoholic | 2 (1.5%) | 2 (1.5%) | |
| Current alcoholic | 11 (8.5%) | 5 (3.8%) | |
| **Hemoglobin (g/dL)** | 12.1 ± 2.5 | 12.0 ± 2.3 | 0.667 |
| **Albumin (g/dL)** | 4.0 ± 0.5 | 3.9 ± 0.5 | 0.067 |
| **Cancer location** | | | 0.323 |
| Cecum | 5 (3.8%) | 2 (1.5%) | |
| Appendix | 1 (0.8%) | 0 | |
| Ascending colon | 20 (15.4%) | 28 (21.5%) | |
| Hepatic flexure | 6 (4.6%) | 9 (6.9%) | |
| Transverse colon | 6 (4.6%) | 12 (9.2%) | |
| Splenic flexure | 3 (2.3%) | 2 (1.5%) | |
| Descending colon | 2 (1.5%) | 5 (3.8%) | |
| Sigmoid colon | 28 (21.5%) | 29 (22.3%) | |
| Rectosigmoid | 10 (7.7%) | 4 (3.1%) | |
| Rectum | 48 (36.9%) | 38 (29.2%) | |
| Double primary | 1 (0.8%) | 1 (0.8%) | |
| **Obstruction** | | | 0.334 |
| None | 96 (73.8%) | 104 (80.0%) | |
| Partial | 19 (14.6%) | 14 (10.8%) | |
| Stent insertion | 13 (10.0%) | 10 (7.7%) | |
| Complete obstruction | 2 (1.5%) | 2 (1.5%) | |
| **Perforation** | | | 0.518 |
| None | 121 (93.1%) | 123 (94.6%) | |
| Microperforation | 7 (5.4%) | 6 (4.6%) | |
| Sealed-off perforation | 2 (1.5%) | 1 (0.8%) | |

(*Continued*)

**Table 2.** (Continued)

| | Control group (n = 130) | Collatamp sponge group (n = 130) | p-value |
|---|---|---|---|
| **Operation type** | | | 0.500 |
| Laparoscopic | 124 (95.4%) | 127 (97.7%) | |
| Robotic | 6 (4.6%) | 3 (2.3%) | |
| **Emergency operation** | 5 (3.8%) | 4 (3.1%) | >0.999 |
| **Transfusion** | | | |
| Preoperative | 13 (10.0%) | 13 (10.0%) | >0.999 |
| Intraoperative | 10 (7.7%) | 9 (6.9%) | 0.812 |
| **Operation methods** | | | 0.185 |
| Right hemicolectomy | 34 (26.2%) | 51 (39.2%) | |
| Left hemicolectomy | 6 (4.6%) | 6 (4.6%) | |
| Anterior resection | 49 (37.7%) | 33 (25.4%) | |
| Low anterior resection | 40 (30.8%) | 38 (29.2%) | |
| Total or subtotal colectomy | 1 (0.8%) | 2 (1.5%) | |
| **Combined resection** | | | 0.294 |
| No | 125 (96.2%) | 122 (93.8%) | |
| Liver | 1 (0.8%) | 2 (1.5%) | |
| Uterus | 1 (0.8%) | 0 | |
| Small bowel | 1 (0.8%) | 2 (1.5%) | |
| Stomach | 1 (0.8%) | 0 | |
| Ovary | 0 | 3 (2.3%) | |
| Spleen | 1 (0.8%) | 0 | |
| Bladder | 0 | 1 (0.8%) | |
| **Operation time (min)** | 193.2 ± 60.9 | 179.7 ± 66.2 | 0.089 |
| **Surgical site infection** | 3 (2.3%) | 2 (1.5%) | >0.999 |
| **Clavien–Dindo classification** | | | 0.546 |
| 0 | 108 (83.1%) | 112 (86.2%) | |
| 1 | 4 (3.1%) | 4 (3.1%) | |
| 2 | 13 (10.0%) | 10 (7.7%) | |
| 3a | 4 (3.1%) | 2 (1.5%) | |
| 3b | 1 (0.8%) | 2 (1.5%) | |
| **Hospital stay (days)** | 6.7 ± 3.7 | 6.5 ± 2.4 | 0.568 |
| **Stage** | | | 0.607 |
| 0 (Tis) | 4 (3.1%) | 2 (1.5%) | |
| 1 | 31 (23.8%) | 31 (23.8%) | |
| 2 | 29 (22.3%) | 27 (20.8%) | |
| 3 | 46 (35.4%) | 49 (37.7%) | |
| 4 | 20 (15.4%) | 21 (16.2%) | |
| **T stage** | | | 0.623 |
| Tis | 4 (3.1%) | 2 (1.5%) | |
| 1 | 17 (13.1%) | 20 (15.4%) | |
| 2 | 17 (13.1%) | 15 (11.5%) | |
| 3 | 74 (56.9%) | 72 (55.4%) | |
| 4a | 17 (13.1%) | 17 (13.1%) | |
| 4b | 1 (0.8%) | 4 (3.1%) | |
| **N stage** | | | 0.812 |
| 0 | 67 (51.5%) | 64 (49.2%) | |
| 1 | 34 (26.2%) | 43 (33.1%) | |
| 2 | 29 (22.3%) | 23 (17.7%) | |

*(Continued)*

**Table 2.** (Continued)

|  | Control group (n = 130) | Collatamp sponge group (n = 130) | p-value |
|---|---|---|---|
| **M stage** |  |  | 0.865 |
| **0** | 110 (84.6%) | 109 (83.8%) |  |
| **1** | 20 (15.4%) | 21 (16.2%) |  |
| **Lymphatic invasion** | 35 (27.6%) | 68 (52.3%) | <0.001 |
| **Venous invasion** | 52 (40.9%) | 46 (35.4%) | 0.359 |
| **Perineural invasion** | 44 (34.6%) | 48 (36.9%) | 0.703 |
| **Adjuvant chemotherapy** | 70 (53.8%) | 67 (51.5%) | 0.709 |

ASA, American Society of Anesthesiologists; BMI, body mass index

A subgroup analysis of patients was performed with the factors that were statistically significant in the multivariable analysis. This analysis revealed that in patients with a BMI >25 kg/m$^2$, the frequency of SSI was lower in the Collatamp group [2.2% (1/84 patients)] than in the control group (7.1% (4/56)) (OR, 0.29; 95% CI, 0.03–2.68; p = 0.375). In the right hemicolectomy group, the Collatamp group [3.9% (2/51 patients)] also exhibited fewer SSIs than the control group [11.1% (5/45 patients)] (OR, 0.33; 95% CI, 0.06–1.77; p = 0.247). However, the differences in these subgroups were not significant (Table 5).

## Discussion

In our study, we found that applying the Collatamp® G to the subcutaneous layer of the specimen extraction site did not affect the incidence of SSI. Therefore, no statistically significant difference in the length of stay and oncological results was observed between the two groups. Furthermore, the frequency of SSI in the Collatamp group was lower than that in the control group in a subgroup analysis of factors that affected the occurrence of SSI; however, the difference was not statistically significant.

SSI increases hospital stays and medical costs [11]. Furthermore, it can also affect a patient's quality of life [12]. Efforts are needed to reduce the incidence of SSIs in cancer patients are needed as they can also affect survival [13–15]. In particular, one study showed that in patients with colon cancer, the 5-year disease-free survival rate was significantly lower in patients with SSI (83%) than in those without (87%) [16]. Thus, various efforts to reduce SSIs are required.

To date, several reports have been published regarding the effects of the gentamicin-collagen sponge. In 2013, in a meta-analysis on the effect of gentamicin-collagen implants on SSIs in all surgical fields, Chang et al. [3] reported that the OR was 0.51 (95% CI, 0.33–0.77; p = 0.001). In another meta-analysis [2], the investigators reported that the RR in four randomized controlled trials (RCT) was 0.61 (95% CI, 0.39–0.98; p = 0.04) in reducing the sternal wound infection after inserting a gentamicin-collagen sponge into the sternal wound after heart surgery.

Various studies have also used investigated the gentamicin-collage sponge in colorectal surgery [4, 17, 18]. A large-scale RCT [4] that involved all colorectal operations performed in 54 countries failed to prove the hypothesis that the gentamicin-collagen sponge would reduce the frequency of SSI. Similarly, an RCT [18] that included 291 patients who underwent laparoscopic colorectal surgery at a single center also failed to prove the efficacy of the gentamicin-collagen sponge. However, in a recent meta-analysis [5], a sensitivity analysis of abdominal wounds revealed that a gentamicin-collagen sponge could reduce SSI in colorectal surgery (RR, 0.38; 95% CI, 0.20–0.72).

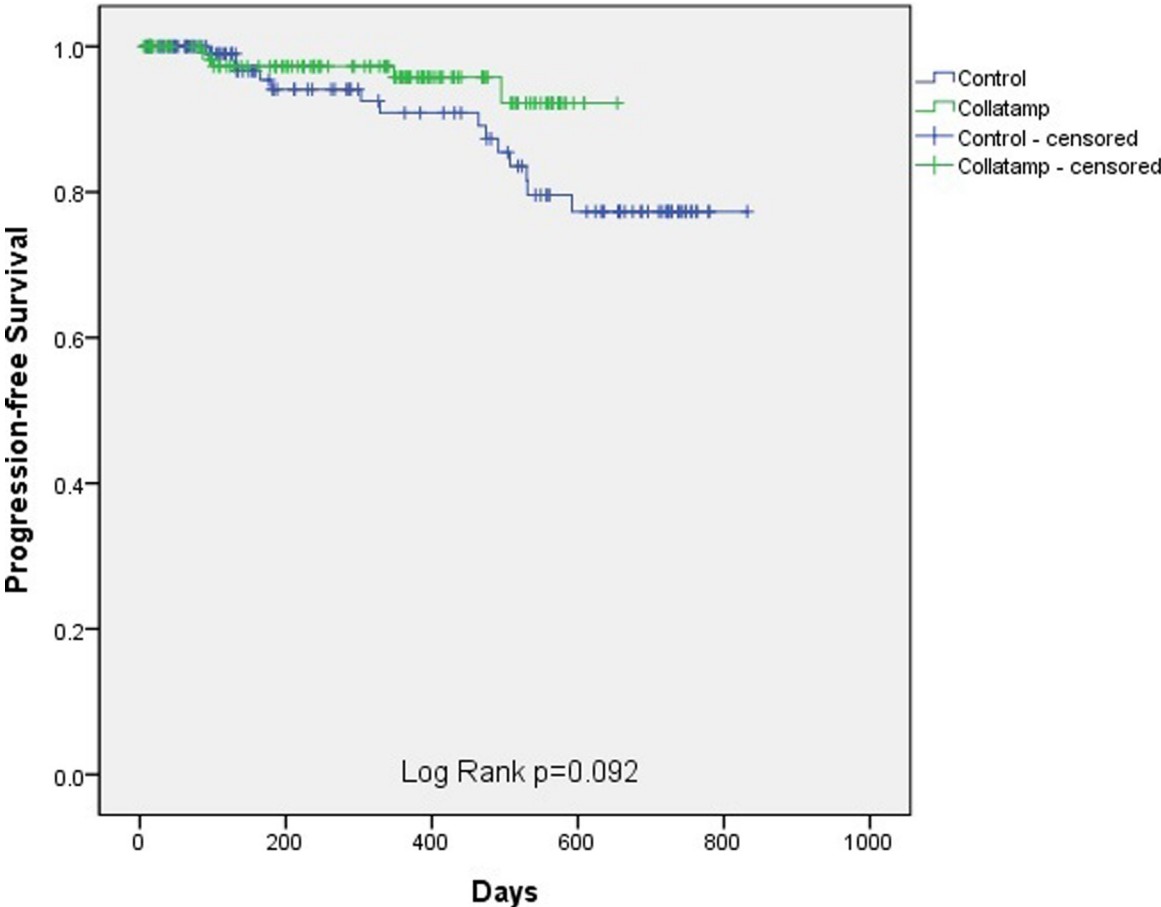

**Fig 2. Progression-free survival between the Collatamp group and the control group.** There was no significant difference in the estimated 2-year progression-free survival rate between the two groups (log-rank p-value = 0.092).

With regard to this debate, we conducted a study to investigate whether the gentamicin-collagen sponge could reduce SSIs, which may affect the oncologic outcome of colon and rectal cancer; however, our study also failed to prove its efficacy.

The reason we could not demonstrate any difference between the two groups in our study was that the frequency of SSIs was low. The incidence of SSIs reported in large-scale RCTs [19–24] ranges from 3.7% to 8.9%. However, the incidence in our study was only 2.5%. This may be explained by our routine use of wound protectors. Dual-ring wound protectors have been widely used to significantly reduce SSI rates after elective surgery for colorectal cancer [25]. Another possible reason is that the sample size was insufficient for the application of the Collatamp to affect SSI. Even in the subgroup analysis, there was no significant difference between the Collatamp and control groups, which may have been due to the small sample size.

The effect of obesity on SSIs in laparoscopic colon surgery has already been demonstrated in other studies [26–28]. A meta-analysis by He et al. [28] revealed that overweightness was associated with an increased risk of SSI compared to normal weight (OR, 1.56; 95% CI 1.36–1.78; p < 0.001). Unlike in Western countries, the BMI cut-off for defining obesity in Asia-Pacific countries is set at 25 kg/m$^2$ [29]. In fact, in our study, there were only 8 patients with a BMI of 30 kg/m$^2$, none of which developed an SSI. Therefore, we also used the 25 kg/m$^2$ cut-

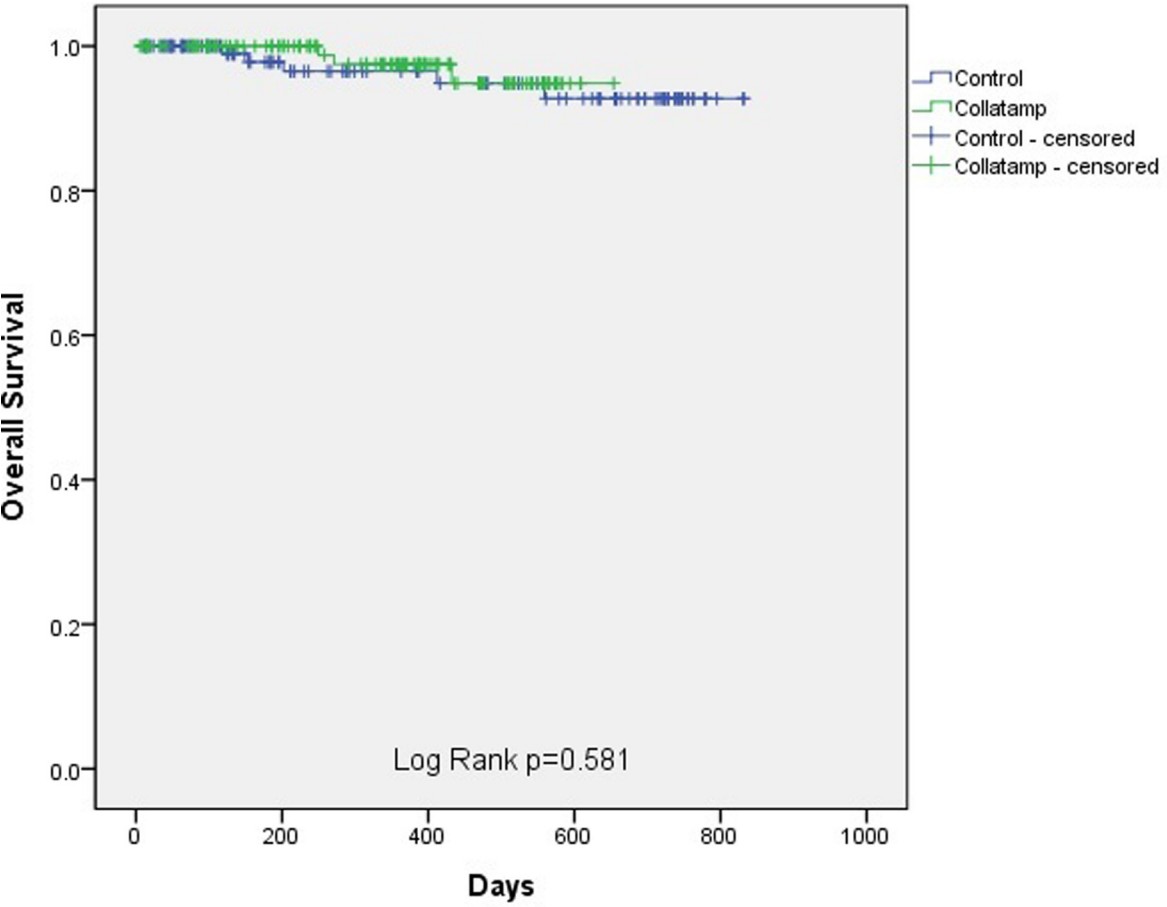

**Fig 3. Overall survival between the Collatamp group and the control group.** There was no significant difference in the estimated 2-year overall survival rate between the two groups (log-rank p-value = 0.581).

off to define obesity. Among obese patients with a BMI of $\geq$25 kg/m$^2$, the incidence of SSI was significantly increased with an OR of 39.0 (95% CI, 1.90–802.21; p = 0.018). In particular, in the subgroup analysis, when the Collatamp was used in the BMI>25 kg/m$^2$ group, the incidence of SSIs were reduced, with an OR of 0.29 (95% CI, 0.03–2.68; p = 0.375), compared with that in patients with a BMI<25 kg/m$^2$; however, this difference was not statistically significant. This finding implied that the Collatamp sponge would be effective in obese patients.

When reviewing the current literature, there are few studies comparing SSIs in laparoscopic procedures such as laparoscopic right hemicolectomy or low anterior resection. Konish et al reported that the frequency of SSI was significantly higher in low anterior resection (9.5%) than in right hemicolectomy (7.5%) (p<0.001) [30]. Similarly, Degrate et al also reported that rectal surgery led to more frequent SSIs than right colonic surgery (17.6% vs 8.0%, p = 0.049) [31]. However, these studies also included open surgery. Our routine use of end-to-side anastomoses during right hemicolectomy may explain the high SSIs rates observed with this procedure in our study, where wound contamination may have occurred due to fecal spillage during insertion and withdrawal of the circular stapler through the incision. Since fecal contamination occurs when the anvil is inserted, the chances of wound contamination are lower in anterior or low anterior resection than in right hemicolectomy, which requires inserting and removing a circular stapler.

**Table 3. Univariable analysis for factors that affect surgical site infection in the overall cohort (n = 310).**

|  | No SSI (n = 302) | SSI (n = 8) | p-value |
|---|---|---|---|
| **Age (y)** | 67.2 ± 11.5 | 71.3 ± 18.4 | 0.220 |
| **Sex** |  |  | 0.715 |
| Male | 186 (97.9%) | 4 (2.1%) |  |
| Female | 116 (96.7%) | 4 (3.3%) |  |
| **BMI (kg/m²)** |  |  | 0.120 |
| <25 | 205 (98.6%) | 3 (1.4%) |  |
| >25 | 97 (95.1%) | 5 (4.9%) |  |
| **Hypertension** |  |  |  |
| None | 126 (95.5%) | 6 (4.5%) | 0.076 |
| Present | 176 (98.9%) | 6 (1.1%) |  |
| **Diabetes** |  |  |  |
| None | 212 (97.7%) | 5 (2.3%) | 0.700 |
| Present | 90 (96.8%) | 3 (3.2%) |  |
| **Cardiac disease** |  |  |  |
| None | 262 (97.0%) | 8 (3.0%) | 0.603 |
| Present | 40 (100%) | 0 |  |
| **Pulmonary disease** |  |  |  |
| None | 275 (97.5%) | 7 (2.5%) | 0.535 |
| Present | 27 (96.4%) | 1 (3.6%) |  |
| **Liver disease** |  |  |  |
| None | 291 (98.3%) | 5 (1.7%) | 0.004 |
| Present | 11 (78.6%) | 3 (21.4%) |  |
| **Cerebrovascular disease** |  |  |  |
| None | 269 (97.1%) | 8 (2.9%) | >0.999 |
| Present | 33 (100%) | 0 |  |
| **Chronic kidney disease** |  |  | 0.417 |
| None | 283 (93.7%) | 7 (87.5%) |  |
| Present | 19 (6.3%) | 1 (12.5%) |  |
| **ASA classification** |  |  | 0.523 |
| 1 | 24 (92.3%) | 2 (7.7%) |  |
| 2 | 218 (98.2%) | 4 (1.8%) |  |
| 3 | 60 (96.8%) | 2 (3.2%) |  |
| **Smoking** |  |  | 0.587 |
| Nonsmoker | 227 (97.0%) | 7 (3.0%) |  |
| Ex-smoker | 32 (100%) | 0 |  |
| Current smoker | 43 (97.7%) | 1 (2.3%) |  |
| **Alcohol** |  |  | 0.983 |
| Nonalcoholic | 263 (97.4%) | 7 (2.6%) |  |
| Ex-alcoholic | 4 (100%) | 0 |  |
| Current alcoholic | 35 (97.2%) | 1 (2.8%) |  |
| **Hemoglobin (g/dL)** |  |  | 0.148 |
| <12 | 135 (95.7%) | 6 (4.3%) |  |
| >12 | 167 (98.8%) | 2 (1.2%) |  |
| **Albumin (g/dL)** |  |  | 0.051 |
| <3.5 | 57 (93.4%) | 4 (6.6%) |  |
| >3.5 | 245 (98.4%) | 4 (1.6%) |  |
| **Obstruction** |  |  | 0.231 |

(*Continued*)

**Table 3.** (Continued)

| | No SSI (n = 302) | SSI (n = 8) | p-value |
|---|---|---|---|
| None | 223 (98.7%) | 3 (1.3%) | |
| Partial | 39 (90.7%) | 4 (9.3%) | |
| Stent insertion | 33 (97.1%) | 1 (2.9%) | |
| Complete obstruction | 7 (100%) | 0 | |
| Perforation | | | <0.001 |
| None | 282 (97.9%) | 6 (2.1%) | |
| Microperforation | 20 (100%) | 0 | |
| Sealed-off perforation | 3 (60.0%) | 2 (40.0%) | |
| Operation type | | | >0.999 |
| Laparoscopic | 287 (97.3%) | 8 (2.7%) | |
| Robotic | 15 (100%) | 0 | |
| Emergency operation | | | 0.274 |
| Elective | 291 (98.4%) | 7 (2.3%) | |
| Emergency | 11 (91.7%) | 1 (8.3%) | |
| Transfusion | | | |
| Preoperative | | | >0.999 |
| None | 272 (97.1%) | 8 (2.9%) | |
| Done | 30 (100%) | 0 | |
| Intraoperative | | | 0.215 |
| None | 270 (97.8%) | 6 (2.2%) | |
| Done | 32 (94.1%) | 2 (5.9%) | |
| Collatamp sponge insertion | | | 0.475 |
| No | 174 (96.7%) | 6 (3.3%) | |
| Yes | 128 (98.5%) | 2 (1.5%) | |
| Operation methods | | | 0.001 |
| Right hemicolectomy | 89 (92.7%) | 7 (7.3%) | |
| Left hemicolectomy | 13 (100%) | 0 | |
| Anterior resection or low anterior resection | 195 (99.5%) | 1 (0.5%) | |
| Total or subtotal colectomy | 5 (100%) | 0 | |
| Combined resection | | | >0.999 |
| None | 284 (97.3%) | 8 (2.7%) | |
| Done | 18 (100%) | 0 | |
| Operation time (min) | 190.6 ± 66.3 | 201.8 ± 44.3 | 0.280 |
| Stage | | | 0.493 |
| 0 (Tis) | 8 (100%) | 0 | |
| 1 | 76 (100%) | 0 | |
| 2 | 65 (95.6%) | 3 (4.4%) | |
| 3 | 106 (95.5%) | 5 (4.5%) | |
| 4 | 47 (100%) | 0 | |

ASA, American Society of Anesthesiologists; BMI, body mass index

## Limitations

The first limitation of our study is its retrospective, single-center design. Because of the retrospective study design, there was a possibility of selection bias. Furthermore, the single-center setting means that our results cannot be generalized. The second limitation is that the sample size was too small to prove the efficacy of the Collatamp in laparoscopic surgery. The

**Table 4. Multivariable analysis for factors that affect surgical site infection.**

|  | Odds ratio | 95% Confidence interval | p-value |
|---|---|---|---|
| BMI >25 (kg/m$^2$) | 39.02 | 1.90–802.21 | 0.018 |
| Albumin <3.5 (g/dL) | 4.75 | 0.60–37.84 | 0.142 |
| Liver disease | 254.76 | 10.43–6222.61 | 0.001 |
| Any type of perforation | 8.86 | 0.51–154.74 | 0.135 |
| Operation methods |  |  | 0.010 |
| AR/LAR | Reference | Reference |  |
| Right hemicolectomy | 36.22 | 2.37–554.63 |  |

AR, anterior resection; BMI, body mass index; LAR, low anterior resection

**Table 5. Subgroup analysis of surgical site infection in high-risk patients.**

|  | Control group | Collatamp sponge group | p-value |
|---|---|---|---|
| BMI (kg/m$^2$) |  |  |  |
| >25 | 4/56 (7.1%) | 1/46 (2.2%) | 0.375 |
| <25 | 2/124 (1.6%) | 1/84 (1.2%) | >0.999 |
| Liver disease |  |  |  |
| None | 4/169 (2.4%) | 1/127 (0.8%) | 0.396 |
| Present | 2/11 (18.2%) | 1/3 (33.3%) | >0.999 |
| Operation methods |  |  |  |
| AR/LAR | 1/125 (0.8%) | 0/71 (0%) | >0.999 |
| Right hemicolectomy | 5/45 (11.1%) | 2/51 (3.9%) | 0.247 |

AR, anterior resection; BMI, body mass index; LAR, low anterior resection

inadequate sample size might lead to insufficient interpretation of the study results. As mentioned previously, the lack of a sufficient sample seems to underlie the inability to confirm statistical significance, even after subgroup analysis. When the sample size was calculated based on the results of our study, 1,134 participants were needed in each group. However, when the sample size was calculated based on the results of the right hemicolectomy group, only 209 patients were needed in each group. Therefore, it would be helpful to conduct multicenter studies focusing on this high-risk group in the future.

## Conclusion

We demonstrated that, in patients with colon or rectal cancer who undergo minimally invasive surgery, applying a gentamicin-collage sponge to the mini-laparotomy wound did not reduce the frequency of SSIs, even in high-risk patients. However, since the possibility of a negative result due to the small sample size cannot be excluded, further multicenter RCTs should be conducted to determine whether the selective use of the gentamicin-collagen sponge may help reduce SSIs in patients at high risk.

## Author Contributions

**Conceptualization:** Kil-yong Lee, Jaeim Lee.

**Data curation:** Kil-yong Lee, Jaeim Lee.

**Formal analysis:** Kil-yong Lee, Jaeim Lee, Youn Young Park, Seong Taek Oh.

**Project administration:** Jaeim Lee.

**Supervision:** Jaeim Lee, Seong Taek Oh.

**Writing – original draft:** Kil-yong Lee, Jaeim Lee, Youn Young Park.

**Writing – review & editing:** Kil-yong Lee, Jaeim Lee, Youn Young Park, Seong Taek Oh.

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
