## [Decision Letter · Decision Letter 0]

10 Jun 2021

PONE-D-21-14973

Use of Gentamicin-Collagen Sponge (Collatamp® G) in laparoscopic colorectal cancer surgery: A propensity score-matched study

PLOS ONE

Dear Dr. Lee,

Thank you for submitting your manuscript to PLOS ONE. After careful consideration, we feel that it has merit but does not fully meet PLOS ONE’s publication criteria as it currently stands. Therefore, we invite you to submit a revised version of the manuscript that addresses the points raised during the review process.

Please address the issues and revise accordingly.

We look forward to receiving your revised manuscript.

Kind regards,

Academic Editor

PLOS ONE

Journal Requirements:

2)  We note that you have indicated that data from this study are available upon request. PLOS only allows data to be available upon request if there are legal or ethical restrictions on sharing data publicly. For information on unacceptable data access restrictions, please see http://journals.plos.org/plosone/s/data-availability#loc-unacceptable-data-access-restrictions.

3) Your ethics statement should only appear in the Methods section of your manuscript. If your ethics statement is written in any section besides the Methods, please delete it from any other section.

Reviewers' comments:

Reviewer's Responses to Questions

**Comments to the Author**

1. Is the manuscript technically sound, and do the data support the conclusions?

Reviewer #1: No

Reviewer #2: Yes

2. Has the statistical analysis been performed appropriately and rigorously? 

Reviewer #1: No

Reviewer #2: Yes

3. Have the authors made all data underlying the findings in their manuscript fully available?

Reviewer #1: No

Reviewer #2: Yes

4. Is the manuscript presented in an intelligible fashion and written in standard English?

Reviewer #1: Yes

Reviewer #2: Yes

5. Review Comments to the Author

Reviewer #1: Authors aim to determine the effectiveness of a gentamicin-collagen sponge to reduce surgical site infections in minimally invasive surgery for colorectal cancer in a single institution between December 1, 2018 and February 28, 2021. After propensity score matching, 130 patients were allocated to each group. No differences in clinical characteristics existed between the two groups. Surgical site infection occurred in 2 (1.5%) patients and 3 (2.3%) patients in the gentamicin-collagen sponge group and the control group, respectively (p<0.999). After analyzing factors that affect surgical site infection, the following factors were found to be statistically significant: body mass index >25 kg/m2 (odds ratio, 39.0; 95% confidence interval, 1.90-802.21; p=0.018), liver disease (odds ratio, 254.8; 95% confidence interval, 10.43-6222.61; p=0.001), and right hemicolectomy (odds ratio, 36.22; 95% confidence interval, 2.37-554.63; p=0.010). In summary, the present study indicated that applying a gentamicin-collagen sponge to mini-laparotomy wound could not reduce the frequency of surgical site infection. Further studies should be conducted regarding whether the selective use of gentamicin-collagen sponges may help reduce surgical site infection in high-risk patients. The results seems informative and appealing; however, there are a lot of criticisms and have several issues that the authors need to address before the manuscript is suitable for publication.

Major Compulsory Revisions:

1. The major flaw of the current study was that authors included only 260 patients divided into 2 groups (130 patients in each group), and the relatively low incidence of SSI (1.5% vs 2.3%); therefore, an estimated sample size analysis is warranted before authors conducting this study to prevent small sample size study bias.

2. In Patients paragraph: The inclusion criteria were laparoscopic or robotic operations for biopsy-proven colorectal cancer and specimen extraction via minilaparotomy wounds. However, the title of the manuscript is Use of Gentamicin-Collagen Sponge (Collatamp® G) in laparoscopic colorectal cancer surgery: A propensity score-matched study. The title needs to be amended.

3. In Statistical analysis paragraph: For multivariable analysis of factors affecting SSI, a logistic regression test was used for factors with a p-value <0.2 in univariable analysis. Authors have to elucidate why set a p-value <0.2 not < 0.05 to reduce false positives.

4. Another major flaw was that authors define body mass index >25 kg/m2 (overweight) as a variable for further analysis, of which it was relatively lower threshold compared to the common definition of obesity was BMI > 30 kg/m2. It would be better to re-analyze by using BMI > 30 kg/m2in the subsequent study.

5. Body mass index >25 kg/m2 (p=0.018), liver disease (p=0.001), and right hemicolectomy (p=0.010) were independent factors affecting SSI. The definitions of liver disease should be clarified with more detailed information, and how about uremia patients undergoing hemodialysis?

6. In Table 2: Total patients with surgical site infection was 5; however, there were 8 SSI in Table 3. The inconsistent data within the manuscript rise a serious concern.

7. Obviously, dual-ring wound protector has been widely used to significantly reduce SSI after elective surgery for colorectal cancer, e.g. Impact of a Dual-Ring Wound Protector on Outcome after Elective Surgery for Colorectal Cancer. J Surg Res 2019;244:136-145. There were no relevant references in the Discussion section.

8. In Discussion, right hemicolectomy was an independent factors affecting SSI should be described in more details with references.

9. In Conclusion paragraph: We demonstrated that in patients with colon cancer or rectal cancer who undergo laparoscopic resection, applying a gentamicin-collage sponge to the mini-laparotomy wound could not reduce the frequency of SSI. However, our findings confirmed that the gentamicin collagen sponge lowered the frequency of SSI among patients at high risk for SSI, although this difference was not statistically significant. The above statements must be extensively revised based on a negative results and no powerful data to demonstrate it.

Minor Essential Revisions:

1. Please correct the typo and grammatical error with an expert good at English-editing.

2. The term of operation name in tables 1-4 that is advised to be substituted with operation methods. .

Reviewer #2: The aim of the manuscript focus on the effect of Gentamicin-Collagen Sponge (Collatamp® G) in the SSI of laparoscopic colorectal surgery. It is interesting and important issue although the result of the study was limited due to small sample size. I have some comments for the authors.

1. What is the definition of the SSI in the study?

2.In methods, for multivariable analysis of factors affecting SSI, a logistic regression test was used for factors with a p-value <0.2 in univariable analysis. But Table 4 didn't show the multivariable analysis result of Hypertension and hemoglobin with p-value< 0.2 in univariable analysis.

3. In discussion, authors thought small sample size was the limitation for less SSI in this study than other previously published data. Please explain why subgroup analysis can compensate for the limitation.

4.Please explain the sentences "In particular, in the subgroup analysis, when the Collatamp was used in the BMI>25 kg/m 2 group, SSI was lowered by an OR of 0.29 (95% CI, 0.03-2.68; p=0.375), compared with that in patients with a BMI<25 kg/m2" in discussion. The data didn't show in result. Table 5 only showed Collatamp decrease SSI rate in BMI>25 kg/m2 (p=0.375).

6. PLOS authors have the option to publish the peer review history of their article (what does this mean?). If published, this will include your full peer review and any attached files.

Reviewer #1: No

Reviewer #2: No

---

## [Author Response · Author response to Decision Letter 0]

16 Jun 2021

Reviewer #1: Authors aim to determine the effectiveness of a gentamicin-collagen sponge to reduce surgical site infections in minimally invasive surgery for colorectal cancer in a single institution between December 1, 2018 and February 28, 2021. After propensity score matching, 130 patients were allocated to each group. No differences in clinical characteristics existed between the two groups. Surgical site infection occurred in 2 (1.5%) patients and 3 (2.3%) patients in the gentamicin-collagen sponge group and the control group, respectively (p<0.999). After analyzing factors that affect surgical site infection, the following factors were found to be statistically significant: body mass index >25 kg/m2 (odds ratio, 39.0; 95% confidence interval, 1.90-802.21; p=0.018), liver disease (odds ratio, 254.8; 95% confidence interval, 10.43-6222.61; p=0.001), and right hemicolectomy (odds ratio, 36.22; 95% confidence interval, 2.37-554.63; p=0.010). In summary, the present study indicated that applying a gentamicin-collagen sponge to mini-laparotomy wound could not reduce the frequency of surgical site infection. Further studies should be conducted regarding whether the selective use of gentamicin-collagen sponges may help reduce surgical site infection in high-risk patients. The results seem informative and appealing; however, there are a lot of criticisms and have several issues that the authors need to address before the manuscript is suitable for publication.

Major Compulsory Revisions:

1. The major flaw of the current study was that authors included only 260 patients divided into 2 groups (130 patients in each group), and the relatively low incidence of SSI (1.5% vs 2.3%); therefore, an estimated sample size analysis is warranted before authors conducting this study to prevent small sample size study bias.

 We sincerely appreciate the reviewer’s valuable comments. As mentioned in the discussion of the study’s limitations, the sample size required was calculated at about 1134 patients in each group, which is practically impossible to achieve with a single center retrospective design. In addition, since the purpose of this study was to conduct a randomized control study using Collatamp, a subsequent multicenter study will be conducted based on this study.

2. In Patients paragraph: The inclusion criteria were laparoscopic or robotic operations for biopsy-proven colorectal cancer and specimen extraction via minilaparotomy wounds. However, the title of the manuscript is Use of Gentamicin-Collagen Sponge (Collatamp® G) in laparoscopic colorectal cancer surgery: A propensity score-matched study. The title needs to be amended.

In accordance with the reviewer’s comment, we have changed the title.

Original:

Use of Gentamicin-Collagen Sponge (Collatamp® G) in laparoscopic colorectal cancer surgery: A propensity score-matched study

Revised:

Use of Gentamicin-Collagen Sponge (Collatamp® G) in minimally invasive colorectal cancer surgery: A propensity score-matched study

3. In Statistical analysis paragraph: For multivariable analysis of factors affecting SSI, a logistic regression test was used for factors with a p-value <0.2 in univariable analysis. Authors have to elucidate why set a p-value <0.2 not < 0.05 to reduce false positives.

We adopted this approach since in previous studies (1,2) multivariable analysis was also performed using variables with p <0.2 in the univariable analysis. We have clarified this in the revised Methods section.

References

1. Nakamura T, Sato T, Takayama Y, Naito M, Yamanashi T, Miura H, et al. Risk Factors for Surgical Site Infection after Laparoscopic Surgery for Colon Cancer. Surg Infect (Larchmt). 2016;17(4):454-8.

2. Hou TY, Gan HQ, Zhou JF, Gong YJ, Li LY, Zhang XQ, et al. Incidence of and risk factors for surgical site infection after colorectal surgery: A multiple-center prospective study of 3,663 consecutive patients in China. Int J Infect Dis. 2020;96:676-81.

4. Another major flaw was that authors define body mass index >25 kg/m2 (overweight) as a variable for further analysis, of which it was relatively lower threshold compared to the common definition of obesity was BMI > 30 kg/m2. It would be better to re-analyze by using BMI > 30 kg/m2in the subsequent study.

The cut-off for obesity in Asia-Pacific countries is different than that in Western countries, being only 25 kg/m2 (1). In fact, in our study, there were only 8 patients with a BMI of 30 kg/m2, none of which developed an SSI. Therefore, we also used the 25 kg/m2 cut-off to define obesity. We have addressed this issue in the revised Discussion.

Reference

1. Pan WH, Yeh WT. How to define obesity? Evidence-based multiple action points for public awareness, screening, and treatment: an extension of Asian-Pacific recommendations. Asia Pac J Clin Nutr. 2008;17(3):370-4.

5. Body mass index >25 kg/m2 (p=0.018), liver disease (p=0.001), and right hemicolectomy (p=0.010) were independent factors affecting SSI. The definitions of liver disease should be clarified with more detailed information, and how about uremia patients undergoing hemodialysis?

Liver disease was defined as hepatitis B, hepatitis C, or any form of liver cirrhosis. We have added this to the Methods section.

In accordance with the reviewer’s comment, we performed a re-analysis to include data regarding chronic kidney disease. There was no difference between the control and Collatamp groups. Furthermore, chronic kidney disease was not a risk factor for surgical site infection. We have added these results to the tables. 

6. In Table 2: Total patients with surgical site infection was 5; however, there were 8 SSI in Table 3. The inconsistent data within the manuscript rise a serious concern.

Table 2 shows the patients after propensity score matching and Table 3 shows all patients before matching; therefore, the numbers are different. To avoid confusion, we have revised the title of Table 3.

7. Obviously, dual-ring wound protector has been widely used to significantly reduce SSI after elective surgery for colorectal cancer, e.g. Impact of a Dual-Ring Wound Protector on Outcome after Elective Surgery for Colorectal Cancer. J Surg Res 2019;244:136-145. There were no relevant references in the Discussion section.

Indeed, wound protectors were used for all mini-lap wounds in our study, which we believe may have resulted in fewer surgical site infections. We have added this point to the revised Discussion.

8. In Discussion, right hemicolectomy was an independent factors affecting SSI should be described in more details with references.

Thank you for your comment. When reviewing the current literature, there are few studies comparing SSIs in laparoscopic procedures such as laparoscopic right hemicolectomy or low anterior resection. Konish et al reported that the frequency of SSIs was significantly higher in low anterior resection (9.5%) than in right hemicolectomy (7.5%) (p<0.001). (30) Similarly, Degrate et al also reported that rectal surgery led to more frequent SSI rates than right colonic surgery (17.6% vs 8.0%, p=0.049)(31). However, these studies also included open surgery. Our routine use of end-to-side anastomoses during right hemicolectomy may explain the high SSIs rates observed with this procedure in our study, where wound contamination may have occurred due to fecal spillage during insertion and withdrawal of the circular stapler through the incision. Since fecal contamination occurs when the anvil is inserted, the chances of wound contamination are lower in anterior or low anterior resection than in right hemicolectomy, which requires inserting and removing a circular stapler. I have added the information to the discussion now.

References

1. Konishi T, Watanabe T, Kishimoto J, Nagawa H. Elective colon and rectal surgery differ in risk factors for wound infection: results of prospective surveillance. Ann Surg. 2006;244(5):758-63.

2. Degrate L, Garancini M, Misani M, Poli S, Nobili C, Romano F, et al. Right colon, left colon, and rectal surgeries are not similar for surgical site infection development. Analysis of 277 elective and urgent colorectal resections. International Journal of Colorectal Disease. 2011;26(1):61-9.

9. In Conclusion paragraph: We demonstrated that in patients with colon cancer or rectal cancer who undergo laparoscopic resection, applying a gentamicin-collage sponge to the mini-laparotomy wound could not reduce the frequency of SSI. However, our findings confirmed that the gentamicin collagen sponge lowered the frequency of SSI among patients at high risk for SSI, although this difference was not statistically significant. The above statements must be extensively revised based on a negative results and no powerful data to demonstrate it.

We have revised the conclusion according to the reviewer's comment.

Minor Essential Revisions:

1. Please correct the typo and grammatical error with an expert good at English-editing.

The manuscript has been revised by a native English speaker provided through a professional academic editing service. The certificate of editing has been attached.

2. The term of operation name in tables 1-4 that is advised to be substituted with operation methods. .

We have revised the terminology according to the reviewer's suggestion.

Reviewer #2: The aim of the manuscript focus on the effect of Gentamicin-Collagen Sponge (Collatamp® G) in the SSI of laparoscopic colorectal surgery. It is an interesting and important issue although the result of the study was limited due to the small sample size. I have some comments for the authors.

1. What is the definition of the SSI in the study?

 An SSI was defined as a clinically reported infection of the mini-laparotomy wound occurring within 30 days after surgery according to the Center for Disease Control and Prevention (CDC) guidelines [1].

Reference

1. Mangram AJ, Horan TC, Pearson ML, Silver LC, Jarvis WR. Guideline for Prevention of Surgical Site Infection, 1999. Centers for Disease Control and Prevention (CDC) Hospital Infection Control Practices Advisory Committee. Am J Infect Control. 1999;27(2):97-132; quiz 3-4; discussion 96.

2. In methods, for multivariable analysis of factors affecting SSI, a logistic regression test was used for factors with a p-value <0.2 in univariable analysis. But Table 4 didn't show the multivariable analysis result of Hypertension and hemoglobin with p-value< 0.2 in univariable analysis.

 We performed logistic regression with backward stepwise selection of factors with a p-value <0.2 in the univariable analysis. Accordingly, hypertension and hemoglobin were eliminated during this process. We have described our approach in the revised Methods to prevent confusion.

3. In discussion, authors thought small sample size was the limitation for less SSI in this study than other previously published data. Please explain why subgroup analysis can compensate for the limitation.

 Subgroup analysis of groups in which SSS was more common was performed to obtain more meaningful results confirming whether Collatamp affects the incidence SSI. However, as pointed out by the reviewer, the analysis may have produced skewed results. Therefore, we have removed mention of these analyses from the revised manuscript.

4. Please explain the sentences "In particular, in the subgroup analysis, when the Collatamp was used in the BMI>25 kg/m 2 group, SSI was lowered by an OR of 0.29 (95% CI, 0.03-2.68; p=0.375), compared with that in patients with a BMI<25 kg/m2" in discussion. The data didn't show in result. Table 5 only showed Collatamp decrease SSI rate in BMI>25 kg/m2 (p=0.375).

 Since the OR was not statistically significant, we did not mention it in the Results. However, to avoid any confusion, we have added them to the revised Results section.

---

## [Decision Letter · Decision Letter 1]

14 Dec 2021

PONE-D-21-14973R1Use of Gentamicin-Collagen Sponge (Collatamp® G) in minimally invasive colorectal cancer surgery: A propensity score-matched studyPLOS ONE

Dear Dr. Lee,

Thank you for submitting your manuscript to PLOS ONE. After careful consideration, we feel that it has merit but does not fully meet PLOS ONE’s publication criteria as it currently stands. Therefore, we invite you to submit a revised version of the manuscript that addresses the points raised during the review process.

Please revise.

We look forward to receiving your revised manuscript.

Kind regards,

Academic Editor

PLOS ONE

Reviewers' comments:

Reviewer's Responses to Questions

**Comments to the Author**

1. If the authors have adequately addressed your comments raised in a previous round of review and you feel that this manuscript is now acceptable for publication, you may indicate that here to bypass the “Comments to the Author” section, enter your conflict of interest statement in the “Confidential to Editor” section, and submit your "Accept" recommendation.

Reviewer #1: (No Response)

Reviewer #2: All comments have been addressed

2. Is the manuscript technically sound, and do the data support the conclusions?

Reviewer #1: Yes

Reviewer #2: Yes

3. Has the statistical analysis been performed appropriately and rigorously? 

Reviewer #1: No

Reviewer #2: Yes

4. Have the authors made all data underlying the findings in their manuscript fully available?

Reviewer #1: Yes

Reviewer #2: Yes

5. Is the manuscript presented in an intelligible fashion and written in standard English?

Reviewer #1: Yes

Reviewer #2: Yes

6. Review Comments to the Author

Reviewer #1: Major Compulsory Revisions:

1. The major flaw of the current study was that authors included only 260 patients

divided into 2 groups (130 patients in each group), and the relatively low incidence of

SSI (1.5% vs 2.3%); therefore, an estimated sample size analysis is warranted before

authors conducting this study to prevent small sample size study bias.

Reply: We sincerely appreciate the reviewer’s valuable comments. As mentioned in the

discussion of the study’s limitations, the sample size required was calculated at about

1134 patients in each group, which is practically impossible to achieve with a single

center retrospective design. In addition, since the purpose of this study was to conduct

a randomized control study using Collatamp, a subsequent multicenter study will be

conducted based on this study.

Query: Authors should mention in the limitations as the inadequate sample size might lead to min-interpretation of results.

Conclusion paragraph: We demonstrated that, in patients with colon or rectal cancer who undergo laparoscopic

resection, applying a gentamicin-collage sponge to the mini-laparotomy wound did not reduce

the frequency of SSIs, even in high-risk patients.

Authors were suggested to correct "who undergo laparoscopic resection" to "who undergo minimally invasive surgery".

Reviewer #2: The authors have responded to all previous comments appropriately. Small sample size is still be concerned.

7. PLOS authors have the option to publish the peer review history of their article (what does this mean?). If published, this will include your full peer review and any attached files.

Reviewer #1: No

Reviewer #2: No

---

## [Author Response · Author response to Decision Letter 1]

16 Dec 2021

Reviewer #1: Major Compulsory Revisions:

1. Query: Authors should mention in the limitations as the inadequate sample size might lead to min-interpretation of results.

Response: We appreciate the reviewer’s thoughtful comments. Accordingly, we have presented this as a limitation

2. Conclusion paragraph: We demonstrated that, in patients with colon or rectal cancer who undergo laparoscopic resection, applying a gentamicin-collage sponge to the mini-laparotomy wound did not reduce the frequency of SSIs, even in high-risk patients.

Authors were suggested to correct "who undergo laparoscopic resection" to "who undergo minimally invasive surgery".

Response: Accordingly, we have revised the manuscript to reflect your suggestion.

---

## [Decision Letter · Decision Letter 2]

14 Feb 2022

Use of Gentamicin-Collagen Sponge (Collatamp® G) in minimally invasive colorectal cancer surgery: A propensity score-matched study

PONE-D-21-14973R2

Dear Dr. Lee,

We’re pleased to inform you that your manuscript has been judged scientifically suitable for publication and will be formally accepted for publication once it meets all outstanding technical requirements.

Kind regards,

Academic Editor

PLOS ONE

Additional Editor Comments (optional):

Reviewers' comments:

Reviewer's Responses to Questions

**Comments to the Author**

1. If the authors have adequately addressed your comments raised in a previous round of review and you feel that this manuscript is now acceptable for publication, you may indicate that here to bypass the “Comments to the Author” section, enter your conflict of interest statement in the “Confidential to Editor” section, and submit your "Accept" recommendation.

Reviewer #1: All comments have been addressed

Reviewer #2: All comments have been addressed

2. Is the manuscript technically sound, and do the data support the conclusions?

Reviewer #1: Yes

Reviewer #2: Yes

3. Has the statistical analysis been performed appropriately and rigorously? 

Reviewer #1: Yes

Reviewer #2: Yes

4. Have the authors made all data underlying the findings in their manuscript fully available?

Reviewer #1: Yes

Reviewer #2: Yes

5. Is the manuscript presented in an intelligible fashion and written in standard English?

Reviewer #1: Yes

Reviewer #2: Yes

6. Review Comments to the Author

Reviewer #1: (No Response)

Reviewer #2: (No Response)

7. PLOS authors have the option to publish the peer review history of their article (what does this mean?). If published, this will include your full peer review and any attached files.

Reviewer #1: No

Reviewer #2: No

---

## [Editor Report · Acceptance letter]

18 Mar 2022

PONE-D-21-14973R2 

Use of Gentamicin-Collagen Sponge (Collatamp^®^ G) in minimally invasive colorectal cancer surgery: A propensity score-matched study 

Dear Dr. Lee:

I'm pleased to inform you that your manuscript has been deemed suitable for publication in PLOS ONE. Congratulations! Your manuscript is now with our production department. 

Kind regards, 

on behalf of

Dr. Robert Jeenchen Chen 

Academic Editor

PLOS ONE